# Update on Prevalence of Pain in Patients with Cancer 2022: A Systematic Literature Review and Meta-Analysis

**DOI:** 10.3390/cancers15030591

**Published:** 2023-01-18

**Authors:** Rolf A. H. Snijders, Linda Brom, Maurice Theunissen, Marieke H. J. van den Beuken-van Everdingen

**Affiliations:** 1Netherlands Comprehensive Cancer Organisation (IKNL), Department of Research & Development, 3511 DT Utrecht, The Netherlands; 2Netherlands Association for Palliative Care (PZNL), 3511 DT Utrecht, The Netherlands; 3Centre of Expertise for Palliative Care, Maastricht University Medical Centre+ (MUMC+), 6229 HX Maastricht, The Netherlands; 4Department of Anaesthesiology and Pain Management, Maastricht University Medical Centre+ (MUMC+), 6229 HX Maastricht, The Netherlands

**Keywords:** cancer pain, prevalence, systematic review, meta-analysis, meta-regression

## Abstract

**Simple Summary:**

Pain associated with cancer diagnoses is a serious concern and one of the most common symptoms reported by cancer patients. The insufficient relief of cancer pain can have a major impact on patients’ quality of life. Recent developments in oncology such as new pain management guidelines, drugs and treatment strategies may have had a positive effect on the prevalence and severity of pain. Therefore, the aim of this systematic literature review was to assess the prevalence of pain and pain severity in cancer patients throughout all phases of treatment in the 2014–2021 literature period. Our results show a decline in both the prevalence and severity of cancer pain, compared to previous research. Nevertheless, with 44.5% of cancer patients still experiencing pain, the prevalence remains high, emphasizing the need for ongoing attention regarding the management of cancer pain.

**Abstract:**

Experiencing pain and insufficient relief can be devastating and negatively affect a patient’s quality of life. Developments in oncology such as new treatments and adjusted pain management guidelines may have influenced the prevalence of cancer pain and severity in patients. This review aims to provide an overview of the prevalence and severity of pain in cancer patients in the 2014–2021 literature period. A systematic literature search was performed using the databases PubMed, Embase, CINAHL, and Cochrane. Titles and abstracts were screened, and full texts were evaluated and assessed on methodological quality. A meta-analysis was performed on the pooled prevalence and severity rates. A meta-regression analysis was used to explore differences between treatment groups. We identified 10,637 studies, of which 444 studies were included. The overall prevalence of pain was 44.5%. Moderate to severe pain was experienced by 30.6% of the patients, a lower proportion compared to previous research. Pain experienced by cancer survivors was significantly lower compared to most treatment groups. Our results imply that both the prevalence of pain and pain severity declined in the past decade. Increased attention to the assessment and management of pain might have fostered the decline in the prevalence and severity of pain.

## 1. Introduction

Each year, more than ten million people worldwide are diagnosed with cancer [1]. Pain associated with cancer diagnoses is a serious concern and one of the most common symptoms reported by cancer patients. Experiencing pain and insufficient relief can be devastating and negatively affect a patient’s performance status and emotional well-being, leading to increased anxiety, anger, feelings of depression and even cognitive dysfunction, reducing a patient’s quality of life [1,2,3,4,5,6].

Despite increased attention to the assessment and management of pain in cancer patients, previous research concluded that no major advances have been made in the management of cancer pain in 50 years [7,8]. A recent systematic literature review on the prevalence of pain during cancer treatment by Evenepoel et al. (2022) showed that pain during cancer treatment remains high during and up to three months after curative cancer treatment [9].

Literature indicates several reasons for the lack of improvement in the prevalence of cancer pain. Common professional barriers include a lack of knowledge and skills, clinicians’ reluctance to prescribe opioids, and poor pain assessment [10]. Patient-related barriers include patients’ reluctance to discuss pain with clinicians, patients’ reluctance to receive treatment for their pain, adherence to analgesic prescriptions, limited knowledge on the assessment of undertreatment, and cognitive and psychological factors of patients such as depression [11,12]. A systematic literature review by Makhlouf and colleagues (2020) found similar attitudinal barriers to effective cancer pain management among professionals and patients, including fear of medication addiction, tolerance of medication, and side effects of opioids [13].

Global trends may have influenced the prevalence of cancer pain. People worldwide are living longer, and ageing populations are associated with increased multimorbidity [14,15,16]. Developments in oncology, including new drugs and treatment strategies, may also have influenced the prevalence of cancer pain and pain severity [17,18,19,20]. Global opioid analgesics use increased between 2015 and 2019, but regional variations exist [21]. This was probably influenced by the response to the opioid epidemic [22,23]. Chen et al. (2022) performed a study on opioid use among cancer patients in the United States and showed a declining trend between 2013 and 2018 [24]. This could potentially lead to further undertreatment of cancer pain. On the other hand, the publication of new pain therapy guidelines may have improved the management of cancer pain [25,26,27].

Many new studies have been published on the prevalence of cancer pain since the 2016 systematic literature review of van den Beuken-van Everdingen et al. The review of Evenepoel et al. (2022) was limited to cancer patients receiving curative treatment, excluding other phases of the disease. Therefore, the aim of this systematic literature review was to assess the prevalence of cancer pain and pain severity in cancer patients throughout all phases of the disease.

## 2. Material and Methods

A systematic literature review was performed in accordance with the recommendations of the “Preferred Reporting Items for Systematic Reviews and Meta-analyses” (PRISMA) guidelines [28].

### 2.1. Search Strategy

A systematic electronic search of the literature published from January 2014 until December 2021 was performed on the 19th of January 2022 using the databases PubMed, Embase, CINAHL, and Cochrane. The search string was inspired by the search strings of van den Beuken-van Everdingen et al. (2016), Evenepoel et al. (2022) and Rietjens et al. (2019) [7,9,29]. Important search terms were “pain”, “symptom”, “prevalence”, and “cancer”. The search string was initially developed for PubMed and later adapted for the other databases: Embase, CINAHL, and Cochrane. More details of the search strategy can be found in Appendix A.

### 2.2. Study Selection

After identification and exclusion of duplicates, one author (R.A.H.S.) screened the titles and abstracts. Full texts of selected studies were examined by all four authors (R.A.H.S., L.B., M.T., and M.H.J.v.d.B.-v.E.) regarding the eligibility criteria. Studies that reported on the prevalence of cancer pain were eligible for inclusion. Inclusion of studies was based on the design, including original studies and secondary analyses of studies in which the patient group had not been included yet, population (studies that included patients ≥18 years, studies reporting on pain, and differentiating between cancer and noncancer patients), setting (including inpatient, outpatient, and palliative care facility (e.g., hospice, palliative care unit, and referred to a palliative care service)), while studies performed in pain clinics/including only patients with pain were excluded (Figure 1). In case of doubt about whether or not to include a study, the study in question was discussed among the authors until consensus was reached.

Corresponding authors of studies that met the inclusion criteria were contacted by e-mail to obtain missing information on the prevalence of pain. If this information remained unclear after a term of two weeks, the respective percentages of pain were estimated based on study figures.

### 2.3. Data Extraction

A standard extraction form was developed for consistent data extraction across the authors. General characteristics were listed for each study and included the author(s), year of publication, continent of origin, study design, method of data collection, aim of the study, and the sample size. Study population characteristics included: ethnicity, gender, age, setting (inpatient, outpatient, patient in palliative care facility), Eastern Cooperative Oncology Group performance status (ECOG), and the type of cancer (head and neck, oesophagus, bronchus/lung, breast, pancreatic, other gastro-intestinal, prostate, other urological, ovary, uterus, other gynaecological, haematological, and other). Studies that included >3 types of cancer were registered as such.

Phases of the disease trajectory were registered and used to categorise studies. Nine groups were defined. Group 1—treatment-naïve cancer patients; Group 2—including cancer patients receiving curative treatment; Group 3—including cancer patients receiving palliative treatment; Group 4—including patients with either curative or palliative treatment, or treatment intent not specified; Group 5—including patients after curative cancer treatment; Group 6—including patients for whom anti-cancer treatment is no longer feasible or wanted; Group 7—including patients in different phases of treatment; Group 8—including patients on anti-cancer treatment (Group 2, Group 3 and Group 4); Group 9—including patients with advanced, metastatic or terminal disease (Group 3 and Group 6).

With regard to the experience of pain, the main percentage of pain (%) in a study was extracted. In cases where a study reported on different types of pain (e.g., headache, abdominal pain, and joint pain) and did not report an overall percentage of pain, the highest percentage was registered. In addition, pain severity was extracted and registered as prevalence (%) of mild, moderate, moderate–severe, and severe pain. In case the severity of pain was not specified in the study and presented with Visual Analogue Scale (VAS) or Numeric Rating Scale (NRS), the rating of Serlin et al. (1995) was applied: none (0), mild (1–4), moderate (5–6), and severe (≥7) [30]. Finally, the pain recall period was extracted (point prevalent, week(s), month(s), year(s)).

### 2.4. Methodological Quality

The quality of the included studies was assessed by all four authors (R.A.H.S., L.B., M.T., M.H.J.v.d.B.-v.E.) using the scoring criteria presented in Table 1. These criteria were also used in the systematic literature review of van den Beuken-van Everdingen et al. (2016) and based on LeBoeuf-Yde and Lauritsen (1995), including criteria developed to assess the methodological quality for prevalence studies [31]. The criteria relate to the representativeness of the study sample (three items); quality of the data (three items); description of the methods and results (three items); and a definition of the prevalence of pain (one item), resulting in a methodological quality score ranging from 0–20 points. Studies were included in this systematic literature review regardless of the methodological quality score.

### 2.5. Data Analyses

Descriptive statistics are presented for both studies with <15 points, and studies with ≥15 points on the methodological quality assessment. Details of the studies scoring <15 points on the methodological quality assessment can be found in Appendix A.

A meta-analysis was performed on the included studies using STATA (version 17.0, StataCorp Texas). The meta-analysis included studies with ≥15 points, a defined sample size, and pain prevalence or proportion of moderate to severe pain reported. Prevalence rates were pooled for each defined patient group with reference to pain, and pain severity (moderate–severe pain). The reciprocal of variance was chosen as a weighting factor to reflect the amount of information that each study contained as it is closely related to the study’s sample size [32]. The 95% Confidence Interval (CI) of each prevalence rate was calculated with the formula: p ± z. √((p(1-p)/n) with p being the pain prevalence rate (p = x/n), with a z-value of 1.96 for a 95% CI. To analyse if the variation in prevalence rates between studies was due to more than chance alone, the grade of heterogeneity was assessed by performing a chi-square test (I^2^). If the heterogeneity test was significant (*p* < 0.05), extra variation was incorporated into the analysis by use of the random-effects model. If the prevalence of pain was 100%, the percentage was set as 99.9% in the analysis to avert statistical issues.

A meta-regression analysis was performed to explore if the pooled pain prevalence rates differed significantly between the patient groups. In addition, meta-regression analyses were also used to explore the association between the prevalence of pain and the type of cancer, ECOG score, age of the population, ethnicity, continent of origin, recall period, setting and method of data collection. The coefficient in the meta-regression analysis indicates how each subgroup differs from the nominated reference group; a negative coefficient in each subgroup suggests a lower prevalence of pain compared to the reference. A number of cancer types were regrouped for the analyses; pancreatic and other gastro-intestinal were regrouped as gastro-intestinal; ovary, uterus and other gynaecological were regrouped as gynaecological cancers.

## 3. Results

### 3.1. Study Selection

The electronic literature search identified 13,483 studies; PubMed yielded 3227 studies, Embase 8340 studies, CINAHL 1200 studies, and Cochrane 716 studies. After the exclusion of duplicates, titles and abstracts of 10,637 studies were screened, of which 1147 were selected for a full text evaluation. The full text assessment resulted in the inclusion of 444 studies. An overview of the reasons for exclusion is presented in Figure 1. Authors (*n* = 6) were e-mailed to obtain missing prevalence data, of which pain prevalence was estimated in five studies because they did not respond.

### 3.2. Study Characteristics

This systematic literature review includes 444 studies, of which 160 studies had <15 points on the methodological quality assessment. Of the 160 studies that scored <15 points on the methodological quality assessment, most were performed in North America (*n* = 46), and the least in Australia/New Zealand (*n* = 6). In 24 studies, the primary objective was to evaluate the prevalence of pain in cancer patients. Most studies included solely gastro-intestinal cancer patients (*n* = 20) and 54 studies included patients of more than three types of cancer (Appendix A).

This systematic literature review included 284 studies with a quality score of ≥15 points (Table 2, Table 3, Table 4, Table 5, Table 6, Table 7 and Table 8). Of the studies with a quality score of ≥15 points, most were performed in Europe (*n* = 97), and the least in Africa (*n* = 7). In 99 studies, the primary objective was to evaluate the prevalence of pain in cancer patients; 185 studies focused on another primary study objective (e.g., quality of life, predictors of depression, fatigue or sleeping problems). Most studies included solely breast cancer patients (*n* = 72), and 91 studies included patients of more than three types of cancer. Two hundred seventy-seven studies of the 284 with a quality score of ≥15 points were included in the meta-analysis.

### 3.3. Prevalence of Pain

The meta-analysis of the pooled pain prevalence rates including all groups (Group 1–Group 7) resulted in an overall pain prevalence of 44.5% (95% CI 41.1–47.9). The pooled pain prevalence rates from the group analyses are presented in Table 9. Forest plots of the meta-analysis on the pooled pain prevalence rates (effect size (ES)) can be found in Appendix A.

Patients in both the group after curative treatment (Group 5) and the treatment-naïve group (Group 1) experienced less pain compared with patients in the palliative treatment group and the group without feasible anti-cancer treatment, respectively Group 3, *p* = 0.000, and *p* = 0.031; and Group 6, *p* = 0.001, and *p* = 0.038. Furthermore, the prevalence of pain was significantly lower in the group after curative treatment (Group 5) compared with the group receiving curative treatment (Group 2, *p* = 0.005), the group receiving all kinds of treatment (Group 4, *p* = 0.001), and the group including patients in different phases of treatment (Group 7, *p* = 0.006).

### 3.4. Pain Severity

The meta-analysis of the pooled pain severity including all treatment groups (Group 1–Group 7) resulted in a prevalence of moderate to severe pain of 30.6% (95% CI 26.9–34.4). The results of the group analyses on the pooled moderate to severe pain prevalence rates are presented in Table 9.

Patients in both the palliative treatment group (Group 3) and the group without feasible anti-cancer treatment (Group 6) experienced more moderate to severe pain compared with patients included in the all-treatments group and the group after curative treatment, respectively; Group 4, *p* = 0.035 and, *p* = 0.023; and Group 5, *p* = 0.006, and *p* = 0.005. In addition, the prevalence of moderate to severe pain was significantly higher in the group including patients in different phases of treatment (Group 7) compared with the group including patients after curative treatment (Group 5, *p* = 0.030).

### 3.5. Determinants of Pain Prevalence

The results of the meta-regression analyses that included all treatment groups (Group 1–Group 7) are presented in Table 10. The prevalence of pain in patients with prostate cancer was significantly lower than in patients with haematological, lung, and breast cancer (*p* = 0.034, *p* = 0.039, and *p* = 0.036 respectively). Age, ethnicity, and performance status (ECOG) were not associated with the overall prevalence of pain (*p* > 0.05). Recall periods of a week(s) (*p* = 0.013) or a month(s) (*p* = 0.001) did result in lower pain rates compared with using point prevalence. Studies that used medical records for data collection showed significantly higher pain rates than studies that used a patient questionnaire (*p* = 0.007). The prevalence of pain in patients who stayed in a palliative care setting was significantly higher compared to the prevalence of pain in outpatients (*p* = 0.040). Studies from South America, Asia and Africa showed significantly higher pain rates compared to studies from Europe (*p* = 0.033, *p* = 0.016 and *p* = 0.000 respectively). Moreover, studies from Africa showed a significantly higher prevalence of pain compared to studies from all other continents, North America (*p* = 0.000), Asia (*p* = 0.000), South America (*p* = 0.014) and Australia/New Zealand (*p* = 0.000).

## 4. Discussion

The current literature on the prevalence of pain and pain severity in patients with cancer shows that both the prevalence of pain and pain severity have declined over the past decade. Pooled pain prevalence rates resulted in an overall prevalence of 44.5%. The treatment group with the lowest prevalence of pain was the group after curative treatment (35.8%). Despite the decline found, the results of this systematic literature review show that the prevalence of pain remains high, especially in advanced, metastatic and terminal cancer patients (54.6%).

The meta-analysis of moderate to severe pain resulted in an overall prevalence of 30.6%. The treatment group with the lowest proportion of moderate to severe pain was the group after curative treatment (22.8%). Moderate to severe pain was most frequently reported in the group including patients without feasible anti-cancer treatment (43.3%).

Some findings deserve attention. First, a slight decrease in the prevalence of pain was found compared with the 2016 systematic literature review by van den Beuken-van Everdingen et al. for the group after curative treatment (35.8% < 39.3%), and for the group during anti-cancer treatment (50.3% < 55.0%) [7]. A more substantial decrease was found for the advanced, metastatic or terminal disease group (54.6% < 66.4%). Our systematic review resulted in a pooled prevalence of 47.8% during curative treatment, which is higher than the 40% found by Evenepoel et al. (2022) [9]. A possible explanation can be found in different inclusion criteria. In contrast to Evenepoel et al. (2022), we did include unsolid tumours and studies conducted in nursing homes. The proportion of cancer types may also have been influential, as Evenepoel et al. (2022) included 12 studies of which ten included breast cancer patients (83%). The proportion of studies that included breast cancer patients in our review was high, but lower compared to the review of Evenepoel et al. (2022).

Second, the prevalence of moderate to severe pain decreased (30.6% < 38.0%) compared with the rate presented in the study by van den Beuken-van Everdingen and colleagues (2016) [7]. In contrast to the 2016 systematic literature review, our meta-analysis did include treatment-naïve patients, and patients after finishing curative treatment. This affects the overall prevalence of moderate to severe pain as this is lower in patients after curative treatment [7].

Several factors may have played a role in the decrease in the prevalence of pain and pain severity in cancer patients. In the literature review of Kwon (2014), the suggestion is made that a multidirectional interdisciplinary approach might be the best way to improve cancer pain management [10]. Therefore, more, and better collaboration between healthcare professionals from different disciplines involved in the patients’ treatment may have led to better management of cancer pain. Shrestha et al. (2022) conducted a systematic literature review and meta-analysis on the effect of pharmacists’ involvement in cancer pain management. Their results showed that pharmacists significantly improved cancer patients’ clinical outcomes related to pain. The direct involvement of pharmacists, or in collaboration with a multidisciplinary oncology team, is highly beneficial for patients [316].

The observed decrease in cancer pain prevalence and pain severity may also be positively impacted by raised awareness and knowledge of healthcare professionals about cancer pain treatment and the publication of new pain therapy guidelines [26,27]. However, there is still room for improvement as the literature indicates that there is a difference in knowledge between healthcare professionals and among professionals from different countries [317,318]. Our results showed that the prevalence of cancer pain was significantly higher in South America, Asia and Africa than in Europe. Silbermann and colleagues have shown that the majority of patients with cancer in low-income countries are undertreated for their pain, partly due to a lack of appropriate education [318]. Furthermore, compared with nurses, physicians have more knowledge about pain management, while both lack knowledge regarding the side effects and pharmacology of opioids [317]. Moreover, a systematic literature review on cancer pain management showed that most oncology nurses have poor knowledge about cancer pain management [319].

Furthermore, in line with our results, previous research on the prevalence of undertreatment of pain in cancer patients indicate that undertreatment of pain decreased over time [320,321,322]. In 2008 and 2014, two systematic literature reviews were published in which the prevalence of undertreatment of pain in cancer patients was 43% and 32%, respectively [320,321]. In 2022, an updated review was published including studies published from 2014 to 2020 that showed a mean weighted prevalence score for undertreatment of 40.2% [322].

Developments in oncology, including new drugs and treatment strategies, may also have had a positive impact on the prevalence of cancer pain and pain severity. Targeted therapy and immunotherapy changed the paradigm of cancer treatment. Compared to conventional chemotherapy, these treatment strategies more subtly reduce and control tumour growth, achieving better survival outcomes and improving patients’ quality of life [323,324]. The literature indicates that patients treated with immunotherapy have less cancer pain and reduced pain severity. For example, the study of Zhou and colleagues showed reduced opioid use and cancer pain scores after adoptive immunotherapy with autologous T-cell infusions [325]. They recommend controlled clinical studies to clarify the relationship between immune cell-mediated immunotherapy and pain relief.

### Strengths and Limitations

A strength of this systematic literature review is its scope, including cancer patients at all stages of the disease. This allowed us to examine differences in the prevalence of pain and pain severity between treatment groups.

However, this systematic literature review has some limitations that should be taken into account when interpreting the results. First, not every study used the criteria of Serlin et al. (1995) for defining mild, moderate or severe pain [30]. This has led to differences in pain severity as the cut-off score for mild and moderate pain differed between these studies. Furthermore, some studies did not describe the severity of pain, or gave an overall prevalence of pain but described multiple types of pain (e.g., headache, abdominal pain, and joint pain). In these cases, the highest percentage of pain was used. Almost no studies mentioned the type of pain (nociceptive, neuropathic, mixed). Another methodological consideration relates to the meta-regression analyses of the determinants of pain. To obtain a more robust statistical analysis with more data, we chose to perform the meta-regression analyses on all studies regardless of treatment group. If a cancer type is over-represented in a treatment group with respectively low or high pain prevalence, this affects the overall prevalence of pain in that cancer type.

Although our analyses did not allow us to draw conclusions about detailed subgroups, this review provides a valuable overview of cancer pain prevalence and pain severity over the past decade.

## 5. Conclusions

In conclusion, this systematic literature review shows that both the prevalence of pain and pain severity have declined in the past decade. The development of new treatment strategies and increased attention on assessment and management of pain may have facilitated the decreases in the prevalence and severity of pain. However, with more than half of the patients experiencing pain, the prevalence in advanced, metastatic and terminal patients remains high. To further decrease the prevalence of pain and pain severity, ongoing attention and improved educational programs on cancer pain management are needed.

## Figures and Tables

**Figure 1 cancers-15-00591-f001:**
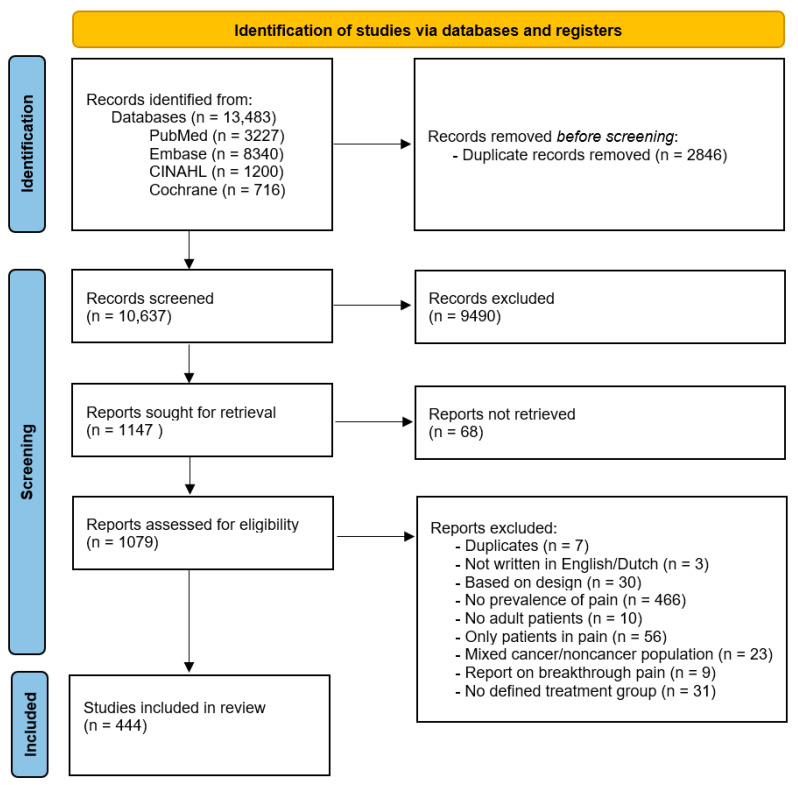
PRISMA flow diagram of study selection.

**Table 1 cancers-15-00591-t001:** Methodological quality criteria for prevalence studies.

A. Representatives of population	1. At least one of the following should apply for the study: an entire target population, randomly selected sample, or sample stated to represent the target population (two points)
	2. At least one of the following: reasons for nonresponse described, non-responders described, comparison of responders and non-responders, or comparison of sample and target population (two points)
	3. Response rate ≥90% (two points) Response rate 70–90% (one point) Response rate ˂70% (zero points)
B. Quality of data	4. Were the primary data from a prevalence study (two points) or were they taken from a survey not specifically designed for the purpose (one point)?
	5. The same mode of data collection should be used for all subjects (two points), if not (zero points).
	6. The data have been collected directly from the patient by means of a validated questionnaire/interview (three points), no validated questionnaire/interview (two points), data have been collected from proxies or retrospectively from medical record (zero points)
C. General description of method and results	7. Description of the target population and setting where patients were found (two points)
	8. Description of stage of disease (one point)
	9. Description of type of cancer, gender, and age: all (two points), 2 of 3 (one point), 1 of 3 (zero points)
	10. Final sample size (one point)
D. Definition of pain prevalence	11. Prevalence recall periods should be stated (one point)

**Table 2 cancers-15-00591-t002:** Pain prevalence in treatment-naïve cancer patients (Group 1) (*n* = 20).

							% Pain					
Study	Quality	Continent ^a^	Setting ^b^	Cancer ^c^	Mean Age	Sample	None	Mild	Moderate	ModSev ^d^	Severe	Overall
Alt-Epping, 2016 [33]	16	2		2	60.6	22	22.7	18.2	45.5	59.1	13.6	77.3
Andersen, 2014 [34]	16	2	2	5		133	60			7		37
Bibby, 2019 [35]	15	2	2	11	64	229						43.7
Efficace, 2015 [36]	16	1,2,3	2	10	70.02	280	45	43.2		11.8		55
Esser, 2017 [37]	15	2	2	10	50.4	239						15
Gjeilo, 2020 [38]	18	2	1,2	4	65.8	264	60					40
Godby–a, 2021 [39] Godby–b, 2021 [39]	18 18	1 1	2 2	6 6	70 70	46 309				75.6 40.7		
Hong, 2015 [40]	15	3	1	6	62.18	165						62.4
Kirchheiner, 2015 [41]	17	2	2	9		50						6.0
Kuon, 2019 [42]	17	2	2	4	63.6	208						34.6
Lunde, 2019 [43]	18	2	2	9	67.2	207						16.9
Ravn Munkvold, 2018 [44]	18	2	1	2		507	48	30	12	22	10	52
Roy, 2016 [45]	16	3	2	4	56.4	36						47.2
Russo, 2018 [46]	17	2	2	2	52.7	527						12.5
Salwey, 2020 [47]	19	2		2		60	26.7	20		53.3		73.3
Tang, 2015 [48]	18	6	1	11	55.1	91	46.8			21.1		63.2
Thomas, 2017 [49]	15	1	2	1	66	105						63
Wang, 2021 [50]	16	3	2	5	50.99	88						4.5
Williams, 2021 [51]	15	1	2	6	70.1	364				45.1		
Yao, 2020 [52]	15	1	2	7		83	69.2	23.1	5.8	7.7	1.9	30.8

^a^ 1 = North America; 2 = Europe; 3 = Asia; 4 = South America; 5 = Africa; 6 = Australia/New Zealand; ^b^ 1 = inpatient; 2 = outpatient; 3 = patient in a palliative care setting; 4 = all; 5 = other; ^c^ 1 = >3 types of cancer; 2 = head and neck; 3 = oesophagus, 4 = bronchus/lung; 5 = breast; 6 = gastro-intestinal; 7 = prostate; 8 = other, urological; 9 = gynaecological; 10 = haematological; 11 = other; ^d^ moderate–severe pain.

**Table 3 cancers-15-00591-t003:** Pain prevalence in patients with curative treatment (Group 2) (*n* = 34).

							% Pain					
Study	Quality	Continent ^a^	Setting ^b^	Cancer ^c^	Mean Age	Sample	None	Mild	Moderate	ModSev ^d^	Severe	Overall
Andersen, 2014 [34]	16	2	2	5		133	20			26		78
Berliere–a, 2021 [53] Berliere–b, 2021 [53]	16 16	2 2	1,2 1,2	5 5	53	32 31						46.0 29.0
Bretschneider, 2016 [54]	16	1	2	9	58.1	152	53.4	29.7	8.8	16.9	8.1	46.6
Browall, 2017 [55]	17	2	2	5	59	124						67
Calderon, 2019 [56]	15	2	2	5	53.2	240						4.2
Choo, 2019 [57]	17	3	2	5	60.35	192						59.9
de Menezes Couceiro, 2014 [58]	16	4	2	5	54	250						44.4
Dylke, 2015 [59]	15	6	2	5	55	157						83
El-Aqoul, 2018 [60]	17	3	1	1	47.6	800	43.6 ^e^			56.4		
Fenlon, 2014 [61]	16	2	2	5	57	455						53
Fjell, 2020 [62]	16	2	2	5	49	150						42.5
Hagiwara–a, 2018 [63] Hagiwara–b, 2018 [63]	19 19	3 3	2 2	6 6		178 176						42.0 41.0
Haryani, 2018 [64]	16	3	2	1	50.51	207		38.2	17.9	22.7	4.8	60.87
Ho, 2015 [65]	15	3	2	5	49	133				24		
Hong, 2014 [66]	16	3	1	1	51.24	1217	88					12
Jarden, 2021 [67]	16	2	1	10	53.1	70						55.2
Jensen, 2018 [68]	17	2	2	9	49	1176	60.6	26.8	9.1	12.6	3.5	39.4
Khan–a, 2017 [69] Khan–b, 2017 [69]	15 15	1 1	2 2	5 5	62 60.5	80 80	36 50	30 20	30 26	30 27	0 1	60 47
Kim, 2016 [70]	16	3	1	5	50.3	1499	8.8 ^e^			91.2		
Kirchheiner, 2015 [41]	17	2	2	9		50						42.0
Kirkham, 2018 [71]	15	1	1	5	50	24						54
Lewis, 2015 [72]	16	6	2	1	61	276						55.0
Lewis, 2021 [73]	18	3	2	2		600						45.2
McFarland, 2018 [74]	15	1	2	5	55.4	125						19.5
Moloney, 2016 [75]	16	6	1	5	54.2	121		24				46.2
Nogueira de Oliveira Martins–a, 2017 [76] Nogueira de Oliveira Martins–b, 2017 [76]	17 17	4 4	2 2	5 5	49.6 49.6	11 11						90.9 72.2
Okamoto, 2018 [77]	17	3	2	5	59	123	48.8	42.3	8.9	14	5.1	51.2
Ribas, 2020 [78]	15	2	2	9	65	109						20.1
Røhrl, 2016 [79]	17	2	2	6	60.7	68						44.1
Shaulov, 2019 [80]	16	1	1,2	10	49.1	318	51.3	8.3	22.9	40.4	17.5	49.2
Tran, 2020 [81]	16	3	1	6	57.7	197						87
Wang, 2021 [50]	16	3	2	5		80						12.5
Xu–a, 2020 [82] Xu–b, 2020 [82] Xu–c, 2020 [82]	15 15 15	3 3 3	2 2 2	5 5 5	51 51 51	235 210 227	66.1 70.5 72.2	20.9 19.5 19.4	11.5 9.5 8.4	12.35 10 8.4	0.85 0.5 0	33.25 29.5 27.8
Yi, 2018 [83]	16	3	2	5	53.56	110	13.6	61.8	20	24.5	4.5	86.4

^a^ 1 = North America; 2 = Europe; 3 = Asia; 4 = South America; 5 = Africa; 6 = Australia/New Zealand; ^b^ 1 = inpatient; 2 = outpatient; 3 = patient in a palliative care setting; 4 = all; 5 = other; ^c^ 1 = >3 types of cancer; 2 = head and neck; 3 = oesophagus, 4 = bronchus/lung; 5 = breast; 6 = gastro-intestinal; 7 = prostate; 8 = other, urological; 9 = gynaecological; 10 = haematological; 11 = other; ^d^ moderate–severe pain; ^e^ none–mild pain.

**Table 4 cancers-15-00591-t004:** Pain prevalence in patients with palliative treatment (Group 3) (*n* = 22).

							% Pain					
Study	Quality	Continent ^a^	Setting ^b^	Cancer ^c^	Mean Age	Sample	None	Mild	Moderate	ModSev ^d^	Severe	Overall
Agarwal, 2020 [84]	15	3	3	1	46.8	110	4.5	24.5	38.2	70.9	32.7	95.5
Al-Zahrani, 2014 [85]	18	3	3	1	56	124	14.5				43.5	85.5
Blank, 2018 [86]	15	1	2	1		13	0	7.7	46.2	92.4	46.2	100 *
Bouché, 2018 [87]	17	2	2	6		151						22.5
Bullock, 2017 [88]	16	1	2	6		40						80
Green, 2015 [89]	17	1	2	7		396			6	8	2	
Iwase, 2015 [90]	16	3	2	1	63.5	183	10.5	25.4	27.6	64.1	36.5	89.5
Jespersen, 2021 [91]	16	2	2	10		92	13.0	23.9	56.5	63.0	6.5	86.9
King, 2018 [92]	15	1,2,3,6	2	9	62.5	903	20.5	38.5	23.1	41	17.9	79.5
Koldenhof–a, 2014 [93] Koldenhof–b, 2014 [93]	16 16	2 2	2 2	6 6	62 66	26 16						23 17
Lavdaniti, 2018 [94]	17	2	1	1	63.8	123	41.5	0	17.1	58.5	41.4	58.5
LeBlanc, 2015 [95]	16	1	2	4	63	97			22.7	32	9.3	
Lechner, 2016 [96]	16	1	1,2	1		62						25.8
Madsen, 2017 [97]	17	2	2	11	63	112						45
McFarland, 2017 [98]	15	1	2	10	57.7	117						21.5
Mercadante–a, 2016 [99] Mercadante–b, 2016 [99] Mercadante–c, 2016 [99] Mercadante–d, 2016 [99]	15 15 15 15	2 2 2 2	4 4 4 4	1 1 1 1	54.5 69.6 79.7 88.8	103 89 116 104	35.0 38.2 42.2 44.2	45.6 34.8 31.9 31.7	15.5 20.2 22.4 20.2	19.4 26.9 25.8 24.0	3.9 6.7 3.4 3.8	65.0 61.7 57.7 55.7
Røhrl, 2016 [79]	17	2	2	6	65.5	52						59.6
Sampogna–a, 2019 [100] Sampogna–b, 2019 [100]	17 17	2 2	2 2	11 11	63.9 63.9	368 65						24 5
Selvy, 2021 [101]	17	2	2	10	66.7	67						59.7
Steel, 2016 [102]	15	1		1	61	261						31.7
Walling, 2015 [103]	15	1		4,6		5422				15		48.2
Zhou–a, 2017 [104]	17	1	2	1	51.4	119				19.3		
Zhou–b, 2017 [104]	17	1	2	1	49.6	42				19.0		
Zhou–c, 2017 [104]	17	1	2	1	50.2	93				43.0		
Zhou–d, 2017 [104]	17	1	2	1	48.7	52				50.0		

^a^ 1 = North America; 2 = Europe; 3 = Asia; 4 = South America; 5 = Africa; 6 = Australia/New Zealand; ^b^ 1 = inpatient; 2 = outpatient; 3 = patient in a palliative care setting; 4 = all; 5 = other; ^c^ 1 = >3 types of cancer; 2 = head and neck; 3 = oesophagus, 4 = bronchus/lung; 5 = breast; 6 = gastro-intestinal; 7 = prostate; 8 = other, urological; 9 = gynaecological; 10 = haematological; 11 = other; ^d^ moderate–severe pain; * 99.9% in meta-analysis.

**Table 5 cancers-15-00591-t005:** Pain prevalence in patients with either curative or palliative treatment, or treatment intent not specified (Group 4) (*n* = 36).

							% Pain					
Study	Quality	Continent ^a^	Setting ^b^	Cancer ^c^	Mean Age	Sample	None	Mild	Moderate	ModSev ^d^	Severe	Overall
Abu-Saad Huijer, 2015 [105]	16	3	2	1	54.32	190						45.3
Bernardes, 2019 [106]	15	6	1,2	1		125	81	10		10		20
Chan, 2015 [107]	16	3		11	57.3	79						25.3
Chen, 2021 [108]	15	3	2	10	41.7	132						57.6
Cheng, 2021 [109]	15	3	2	5		127						21.3
Damm, 2020 [110]	16	2	1,2	6		139		43.9	17.9	18.6	0.7	63
Fujii, 2017 [111]	19	3	2	1	60.8	524	46.2	39.9	7.6	13.9	6.3	53.8
Gosselin, 2016 [112]	15	1	2	6		275	48	24	28	28	0	52
Han, 2019 [113]	15	1	2	1	57.9	399						59.4
Joseph, 2021 [114]	15	5	2	1		347	14.1	21.9	48.7	64	15.3	85.9
Kim, 2015 [115]	15	3	2	1	60.02	167				25.7		92.8
Kuperus, 2021 [116]	15	1	2	8		113	74	14.7	8.8	11.3	2.5	26
Lee–a, 2016 [117] Lee–b, 2016 [117]	20 20	1 1	2 2	5 5	56.0 56.0	358 335		16 ^f^ 31 ^f^				
Li, 2017 [118]	15	3		1	55.36	317	60.25	29.02	5.36	10.41	5.05	39.43
Llamas Ramos, 2016 [119]	15	2	2	1	59.98	246		13.4	24.4	37	12.6	50.4
Matsumoto, 2020 [120]	19	3	2	6		45						55.6
Molassiotis, 2019 [121]	16	2,3	1,2	1	55.15	343						14.2
Moye, 2014 [122]	16	1		2,3,6	64.66	170			21.6	32.4	10.8	
Pearce–a, 2017 [123] Pearce–b, 2017 [123] Pearce–c, 2017 [123]	15 15 15	6 6 6	2 2 2	5 6 4		243 142 56						74 77 77
Pérez, 2015 [124]	17	2	2	1	60.7	358	45.8			26.3		54.2
Pettersson, 2014 [125]	16	2	2	6	65	104						41
Raj, 2014 [126]	19	2	2	1	60	305						48.2
Ritchie–a, 2014 [127] Ritchie–b, 2014 [127] Ritchie–c, 2014 [127] Ritchie–d, 2014 [127]	16 16 16 16	6 6 6 6	2 2 2 2	1 1 1 1	62.1 67.3 72.4 78.8	78 94 76 82						66.7 50.0 59.2 53.7
Salvetti, 2020 [128]	15	4		1	55	107						42.1
Schumacher–a, 2021 [129] Schumacher–b, 2021 [129]	15 15	6 6	2 2	7 7	67.9 67.8	72 43						18.1 23.3
Seven, 2016 [130]	15	2	1	9	59	134	57.3	15.7	20.2	26.9	6.7	42.6
Spoelstra, 2015 [131]	16	1	2	1	65.1	30						63.3
Stamm, 2021 [132]	15	2	2	10	61	47						69
Steffen McLouth, 2020 [133]	16	1		4	62.5	60						41.7
Thiagarajan, 2016 [134]	16	3	1,2	1	52.7	303	15.6	13.8	19.4	23.8	4.4	37.6
Turner, 2014 [135]	15	6	2	1	76.7	385				26		
Unseld, 2021 [136]	15	2	2	1	57.4	846	36.5	43.5	13.6	20	6.4	63.5
van der Baan, 2020 [137]	15	2	1,2	1	58.2	1919						24
Wang, 2014 [138]	16	3		4	58.25	183						71
Yahaya, 2015 [139]	16	3		1	53.4	268						51.1
Zhong, 2017 [140]	17	3	1	1	59.7	517	70.2 ^e^			29.8		

^a^ 1 = North America; 2 = Europe; 3 = Asia; 4 = South America; 5 = Africa; 6 = Australia/New Zealand; ^b^ 1 = inpatient; 2 = outpatient; 3 = patient in a palliative care setting; 4 = all; 5 = other; ^c^ 1 = >3 types of cancer; 2 = head and neck; 3 = oesophagus, 4 = bronchus/lung; 5 = breast; 6 = gastro-intestinal; 7 = prostate; 8 = other, urological; 9 = gynaecological; 10 = haematological; 11 = other; ^d^ moderate–severe pain; ^e^ none–mild pain; ^f^ mild–moderate pain.

**Table 6 cancers-15-00591-t006:** Pain prevalence in patients after curative treatment (Group 5) (*n* = 88).

							% Pain					
Study	Quality	Continent ^a^	Setting ^b^	Cancer ^c^	Mean Age	Sample	None	Mild	Moderate	ModSev ^d^	Severe	Overall
Adams, 2014 [141]	17	2	2	1	70.6	418						15.5
Ahmed, 2014 [142]	17	3	2	5	49.33	73						5.4
Al Maqbali, 2021 [143]	15	3	2	5		133						45.1
Andersen, 2015 [144]	19	2	2	5		475	86					14
Asplund, 2015 [145]	18	2	2	6		545	79				7	21
Baden, 2020 [146]	18	2	2	7		3348						24
Bao, 2018 [147]	18	1	2	5		1280						33.3
Bennedsgaard–a, 2020 [148] Bennedsgaard–b, 2020 [148]	17 17	2 2	2 2	5 6	56.6 68.0	80 52						47.5 45.1
Boehmer–a, 2021 [149] Boehmer–b, 2021 [149]	17 17	1 1	2 2	6 6		116 302						6.1 6.6
Bøhn, 2019 [150]	16	2	2	1	49.1	1088				10		
Bonhof, 2020 [151]	17	2	2	6		477						9
Bovbjerg, 2019 [152]	15	1	2	5	59.4	417						50.6
Bulley, 2014 [153]	15	2	2	5	62.3	595						28.8
Cameron, 2018 [154]	16	1	2	1	26.9	124						54.5
Capelan, 2017 [155]	15	2	2	5		214						29
Chiang, 2019 [156]	18	6	2	5	60.2	201				23	4	55
Cramer, 2018 [2]	18	1	2	2		175	54.9	22	11.6	23.1	11.5	45.1
Cui, 2018 [157]	15	3	2	5	52.4	420						36.2
de Groef, 2017 [158]	16	2	2	5	54.1	147						50
Dieterich–a, 2021 [159] Dieterich–b, 2021 [159]	15 15	2 2	2 2	5 5	60.9 64.4	120 55	60.8 62.3	27.5 20.8	8.4 15	11.7 16.9	3.3 1.9	39.2 37.7
Drury, 2017 [160]	17	2	2	6	66.4	252	64	26	8	10	2	36
Dualé, 2014 [161]	18	2	2	5		442						37.1
Efficace, 2019 [162]	15	2	2	10	52.5	244	51	27		22		49
Engvall, 2021 [163]	16	2	2	5	60.7	646						46.3
Esser, 2017 [37]	15	2	2	10	47.7	102						8
Ezendam, 2014 [164]	17	2	2	9	65	180						31.7
Farrukh, 2020 [165]	17	1	2	10		736					39.4	
Feddern, 2015 [166]	19	2	2	6		1369		13	13	18	5	31
Ferreira, 2019 [167]	18	2	2	5		4262					51	
Gallaway, 2020 [168]	17	1	2	1		12,019						9.5
Gjeilo, 2020 [38]	18	2	2	4		194						55
Gong, 2020 [169]	16	3	2	5	49.3	1983						28.2
Götze–a, 2018 [170] Götze–b, 2018 [170]	16 16	2 2	2 2	1 1	66.3 67.3	660 342						51.2 50.8
Hammer, 2014 [171]	15	1	2	9	63.5	213				22.5		
Hamood–a, 2016 [172] Hamood–b, 2016 [172]	15 15	3 3	2 2	5 5	57 50	54 29	25.9 10.3	9.3 10.3	48.2 34.5	48.2 55.2	0 20.7	57.4 65.5
Hamood, 2018 [173]	17	3	2	5		410			64			74.4
Haviland, 2017 [174]	17	2	2	5	55.1	864						32.3
Henderson–a, 2014 [175] Henderson–b, 2014 [175]	16 16	2 2	2 2	5 5	63 49	138 134	73.2 61.9	17.4 26.1	9.4 10.4	9.4 11.9	0 1.5	26.8 38
Henry, 2020 [176]	16	1		2	63.3	145		16.6	15.9	21.4	5.5	37.9
Hope-Stone, 2015 [177]	15	2	2	11	66.7	179	77					23
Huang, 2020 [178]	15	1	2	1	33.6	1208						18.7
Janah, 2020 [179]	16	2	2	1		4093	36.5					63.5
Jansen, 2018 [180]	15	2	2	2	70	283						8.8
Jardim, 2020 [181]	16	4	2	5	55	151						18.5
Jariwala, 2021 [182]	19	3	2	5	50	212						12
Johannsdottir, 2017 [183]	15	2	2	11		124	56.1	33.3	10.6	10.6	0	43.9
Johannsen, 2015 [184]	17	2	2	5		1905						72.7
Juhl, 2016 [185]	19	2	2	5	63.6	261		13	19.2	25.3	6.1	38.3
Karlson, 2020 [186]	15	1	2	1		10,012	71.4 ^e^		20.5	28.6	8.1	
Kaur, 2018 [187]	15	3	2	5	49.76	230						63.5
Kelada, 2019 [188]	16	6	2	10,11	26.27	404						28.7
Kibar, 2017 [189]	16	2	2	5	52.5	201	68.2					31.8
Kjaer, 2016 [190]	15	2	2	2	64	369				16.0		
Knowlton, 2020 [191]	15	1	2	1		294						24.2
Koehler, 2018 [192]	19	1	2	5	56	36						44.0
Kramer, 2019 [193]	15	5	2	5	60.05	349	26	46	14	28	14	74
Lou, 2021 [194]	16	1	2	2	59.3	77		40 ^f^				
Lunde, 2019 [43]	18	2	2	9	67.2	207						14.9
Lunde, 2020 [195]	19	2	2	9	66.1	140						13.6
Madsen, 2017 [97]	17	2	2	11	63	95						37
Mao, 2018 [196]	15	1	2	5		1103				26.2		
Mertz, 2017 [197]	19	2	2	5		473	67	21		12		33
Miaskowski, 2014 [198]	18	1	2	5	54.98	394	41.6					58.4
Min, 2021 [199]	15	3	2	7	64	111						12.6
Mulrooney, 2019 [200]	19	1	2	10		980	70	23	5	7	2	30
Paek, 2019 [201]	18	3	2	1	62.2	1037	71.9			28.1		28.1
Park, 2018 [202]	17	3	2	11	36.43	144	75					25
Poço Conçalves, 2021 [203]	17	2	2	1	65.33	85	77.6	0	20	22.4	2.4	22.4
Ravn Munkvold, 2018 [44]	18	2	2	2		357	70	19	7	11	4	30
Reilly, 2016 [204]	17	1	2	10	47.6	31						68
Ren, 2021 [205]	17	1	2	2	61.5	505	55	17.7	19.5	26.9	7.4	45
Rogers, 2021 [206]	15	1	2	11	47	248						49
Rosenberg, 2015 [207]	15	1	2	5	55.8	2086						48
Sanchez-Birkhead, 2017 [208]	15	1	2	5	52.6	35				38.3		79
Sanford, 2019 [209]	15	1	2	1		7565	67.5					32.5
Schou Bredal, 2014 [210]	16	2	2	5	56	834		21	17	20	3	41
Selvy, 2020 [211]	16	2	2	6	66.3	406						11.5
Steyaert, 2016 [212]	15	2	2	5	56.5	128						43.8
Tang, 2015 [48]	18	6	1	11	55.1	76	53.9			19.7		46.1
Terkawi, 2017 [213]	16	3	2	2	49.6	102						30
Tonning Olsson, 2021 [214]	16	1	2	1	32.2	2836	61.1			38.9		38.9
Tung, 2019 [215]	18	1	2	6		615				44.87		
van de Luijtgaarden, 2014 [216]	17	2	2	11	45.7	24						93
van Eck, 2020 [217]	15	2	2	11	60	752						14
Variawa, 2016 [218]	18	5	2	5	58.54	92				38.04		
Vuksanovic, 2021 [219]	16	6	2	5		130				34.4		
Wang, 2021 [50]	16	3	2	5		75						4.5
Wilson, 2020 [220]	17	1	1	11		136						16.3

^a^ 1 = North America; 2 = Europe; 3 = Asia; 4 = South America; 5 = Africa; 6 = Australia/New Zealand; ^b^ 1 = inpatient; 2 = outpatient; 3 = patient in a palliative care setting; 4 = all; 5 = other; ^c^ 1 = >3 types of cancer; 2 = head and neck; 3 = oesophagus, 4 = bronchus/lung; 5 = breast; 6 = gastro-intestinal; 7 = prostate; 8 = other, urological; 9 = gynaecological; 10 = haematological; 11 = other; ^d^ moderate–severe pain; ^e^ none–mild pain; ^f^ mild–moderate pain.

**Table 7 cancers-15-00591-t007:** Pain prevalence in patients without feasible anti-cancer treatment (Group 6) (*n* = 17).

							% Pain					
Study	Quality	Continent ^a^	Setting ^b^	Cancer ^c^	Mean Age	Sample	None	Mild	Moderate	ModSev ^d^	Severe	Overall
Aktas, 2014 [221]	17	1	4	1		1000						84
Corli, 2020 [222]	19	2	1,2,4	1	74.2	865						60.5
de la Cruz, 2015 [223]	15	1	4	1		78						51
Drat-Gzubicka, 2021 [224]	18	2	3			76						29
Gupta, 2016 [225]	16	3	1	1	52.49	110	32.73	11.82	22.73	55.46	32.73	67.27
Guthrie, 2019 [226]	15	1	2	1		21,119						48.3
Hui, 2020 [227]	18	1	4	1	60	200						67
Mejin, 2019 [228]	15	3	3	1	57,1	151	18.5	14.6	19.2	66.9	47.7	81.5
Mercadante, 2018 [229]	15	2	3	1	72.1	263					6.9	
Morita–a, 2014 [230] Morita–b, 2014 [230]	16 16	3 3	4 5	1 1	67 68	859 857	42 43	41 40	8.9 8.4	17.2 16	8.3 7.6	58 59
Rojas-Concha, 2020 [231]	17	2	3	1		5449	95.8	0.4	1.8	3.9	2.1	4.2
Seow–a, 2021 [232] Seow–b, 2021 [232]	15 15	1 1	2 1			11,407 15,888				64.9 61.4		
Silvia, 2021 [233]	17	4	3	1		490	49.6	26.5	12.9	23.9	11	50.4
Tofthagen, 2019 [234]	16	1	4	1	72.7	717						68.9
van der Baan, 2020 [137]	15	2	1,2	1	63.4	224						45
Yamagishi, 2014 [235]	16	3	1	1	67	859	84					16
Yanaizumi, 2021 [236]	17	3	2	1		108	6.5	12.9		80.6		93.5

^a^ 1 = North America; 2 = Europe; 3 = Asia; 4 = South America; 5 = Africa; 6 = Australia/New Zealand; ^b^ 1 = inpatient; 2 = outpatient; 3 = patient in a palliative care setting; 4 = all; 5 = other; ^c^ 1 = >3 types of cancer; 2 = head and neck; 3 = oesophagus, 4 = bronchus/lung; 5 = breast; 6 = gastro-intestinal; 7 = prostate; 8 = other, urological; 9 = gynaecological; 10 = haematological; 11 = other; ^d^ moderate–severe pain.

**Table 8 cancers-15-00591-t008:** Pain prevalence including patients in different phases of treatment (Group 7) (*n* = 79).

							% Pain					
Study	Quality	Continent ^a^	Setting ^b^	Cancer ^c^	Mean Age	Sample	None	Mild	Moderate	ModSev ^d^	Severe	Overall
Aguilar, 2017 [237]	16	1	2	2	61.7	23						20.3
Alemayehu, 2018 [238]	18	5	1,2	1	43.5	390	31 ^e^			69		
Armstrong, 2016 [239]	15	1	2	11		621						15.6
Bacorro, 2017 [240]	15	3	1,2	2	51.8	100	53	26	9	21	12	47
Batalini, 2017 [241]	16	4	2	1	59	277						40
Bauml, 2015 [242]	16	1	2	5	60.5	203	78.7 ^e^			21.3		
Beesley, 2016 [243]	15	6	2	6	67	116						32
Bhattacharya, 2019 [244]	16	2	2	5		957						10.5
Bonhof, 2018 [245]	16	2	2	9	64.8	98						37.5
Bouhassira, 2017 [246]	18	2	2	1	61.4	509						28.2
Boyes, 2015 [247]	17	6	2	10		311						12
Braamse–a, 2014 [248] Braamse–b, 2014 [248]	16 16	2 2	2 2	10 10	55.4 57.8	123 125						34.8 32.2
Braga-Neto, 2018 [249]	16	4	2	6		207						83
Bubis, 2020 [250]	15	1	2,4	1		22,650						33
Buckley, 2018 [251]	16	1	1,2	10		82						60
Cho–a, 2019 [252] Cho–b, 2019 [252]	16 16	3 3		2 1	53.7 58.02	708 2581						60.6 44
Clover, 2017 [253]	15	6	2	1	62	9133						17
Daly, 2020 [254]	15	2	5	1		1027						25.3
Davies, 2021 [255]	17	2	3	1		250		6.0	11.6	24.5	12.9	30.5
Davis, 2018 [256]	15	1	2	5	59.7	23,840				35.5	13.2	
de Groef, 2016 [257]	18	2	2	5	60.5	100						50
de Mello, 2020 [258]	16	4	1	6		348						55.61
de Oliveira, 2014 [259]	18	1	2	5		300						36.7
Deshields, 2014 [260]	16	1	2	1	60.6	513						44
Deshields, 2017 [261]	16	1	2	1	60.63	558						45
Dhingra, 2015 [262]	17	1	2	1	62.3	1436						44.9
Donovan, 2014 [263]	18	1	2	9	56	104						31.7
Doubova–a, 2021 [264] Doubova–b, 2021 [264]	19 19	4 4	2 2	5 9		247 165	75.3 ^e^ 67.3 ^e^			24.7 32.7		
Feiten, 2014 [265]	16	2	2	5		730						34
Fokdal, 2018 [266]	18	1,2,3	2	9		914						13.4
Giuliani, 2016 [267]	15	1	2	4		89						21
Götze, 2019 [268]	16	2	2	10	75.6	200						34
Gough, 2017 [269]	16	2		11	59	113				24		77
Guedes, 2018 [270]	16	4	2	5	55.97	103	44.7					55.3
Honkanen, 2021 [271]	16	2	2	5	56.9	121		35.5		14		49.5
Horick, 2018 [272]	18	1	2	1		309						45
Huang, 2018 [273]	17	3	1	6	59	300	39.7	25	25.3	35.5	10	60.3
Hunnicutt–a, 2017 [274] Hunnicutt–b, 2017 [274]	15 15	1 1	4 1	1 1		27,790 31,271	40.0 50.0	38.2 31.6		21.7 18.4		59.9 50.0
Jesdale–a, 2020 [275] Jesdale–b, 2020 [275] Jesdale–c, 2020 [275]	15 15 15	1 1 1	2 2 2			329,150 42,317 12,290						77.8 74.6 75.4
Jewett, 2020 [276]	17	1	2	9	59.9	355						25.1
Karawekpanyawong–a, 2021 [277] Karawekpanyawong–b, 2021 [277]	16 16	3 3	2 2	9 9	52.7 55.8	27 173						23.51 46.3
Khan, 2021 [278]	15	3	2	5		97	75.3					24.7
Kim, 2015 [279]	15	3	2	5	52.1	827	56.5				1.1	43.5
Kim–a, 2017 [280] Kim–b, 2017 [280]	15 15	3 3	2 2	2 2	53.2 43.6	77 81						37.7 27.0
Lefkowits, 2014 [281]	15	1	2	9		305				32		60.1
Liu, 2015 [282]	16	1	5	5	51.7	97						53.6
Loh, 2018 [283]	15	1	2	1		389						49.6
Lu, 2017 [284]	15	3	2	1	52.9	919						49.4
Mavros, 2020 [285]	16	1	5	6		2168						39
Mejdahl, 2015 [286]	15	2	2	5		286				28		
Nakanotani, 2014 [287]	15	3	2	1	57.6	807				17.6		
Nappi, 2021 [288]	15	2	2	5	58.4	175						65.9
Oosterling, 2016 [289]	17	2	2	1		892	77 ^e^			23		
Pandya, 2019 [290]	16	1	1	1	81	359						56
Pinto, 2021 [291]	15	5	1,3	1	46.1	294			33.3	62.6	29.3	83.7
Ramsenthaler, 2016 [292]	15	2	1,2	10	68.4	557		24	34	47.8	13.8	71.5
Rau, 2015 [293]	19	3	2	1	57.47	2075						50.7
Reuter, 2019 [294]	16	1	2	1	61.62	817						35.4
Rosario, 2019 [295]	17	1	1	1	75	33						60
Saarelainen, 2014 [296]	15	6	2	1	76.7	385				26		
Sager, 2020 [297]	18	1	2	1	65.31	123						26.8
Sakurai, 2019 [298]	17	3	2,3	1	64.4	142						66.2
Sampogna, 2020 [299]	16	2	2	11								44.3
Sánchez Sánchez, 2020 [300]	15	2	2	7	70.71	184	60.9	39.1				39.1
Schreiber, 2021 [301]	15	1	2	5	55.5	201						32
Shen, 2017 [302]	16	3	2	1	57	2652	37.4	29.8	23.2	32.8	9.6	62.6
Sloan–a, 2016 [303] Sloan–b, 2016 [303]	16 16	1 1	2 2	4 4	66 66	296 1013						49.0 33.5
Smith, 2019 [304]	15	1	2	5,6	59.7	2257						61.2
Tadele, 2015 [305]	18	5	2	1		388						77.6
Tegegn, 2017 [306]	15	5	1	1		83	8.4	25.3	57.8	66.2	8.4	91.6
Tjong, 2021 [307]	17	1	2	4		13,289				62		
Tung, 2019 [308]	18	1	2	6		2623				61.8		
Valery, 2017 [309]	15	6		1		82						10
van Arsdale, 2016 [310]	16	1	2	9	59	165				18.8		
van den Brekel, 2020 [311]	15	2	1	1	60	2144						28
Vistad, 2018 [312]	16	2		9	61.9	82						56
von Gruenigen, 2018 [313]	17	1	2	9	63.6	65						69
Williams, 2020 [314]	17	1	1	6	70.1	336						20.9
Xu, 2017 [315]	15	3	2	11	43.5	142					30	

^a^ 1 = North America; 2 = Europe; 3 = Asia; 4 = South America; 5 = Africa; 6 = Australia/New Zealand; ^b^ 1 = inpatient; 2 = outpatient; 3 = patient in a palliative care setting; 4 = all; 5 = other; ^c^ 1 = >3 types of cancer; 2 = head and neck; 3 = oesophagus, 4 = bronchus/lung; 5 = breast; 6 = gastro-intestinal; 7 = prostate; 8 = other, urological; 9 = gynaecological; 10 = haematological; 11 = other; ^d^ moderate–severe pain; ^e^ none–mild pain.

**Table 9 cancers-15-00591-t009:** Pooled pain prevalence, and severity rates by treatment group.

	N Studies	N cancer Patients	Overall Pain
Group 1	18	3294	40.3% (95% CI 30.2–50.4)
Group 2	31	7995	47.8% (95% CI 39.4–56.1)
Group 3	19	8719	54.3% (95% CI 42.0–66.6)
Group 4	32	9118	50.3% (95% CI 42.5–58.0)
Group 5	77	59,766	35.8% (95% CI 31.5–40.1)
Group 6	15	33,162	55.2% (95% CI 39.2–71.3)
Group 7	66	508,827	45.0% (95% CI 39.8–50.1)
Group 8	81	25,832	50.3% (95% CI 44.6–56.1)
Group 9	34	41,881	54.6% (95% CI 43.8–65.5)
			Moderate–Severe Pain
Group 1	9	1895	33.1% (95% CI 22.0–44.2)
Group 2	12	5483	27.8% (95% CI 6.7–48.8)
Group 3	11	8057	39.1% (95% CI 30.0–48.1)
Group 4	16	4966	25.2% (95% CI 19.0–31.4)
Group 5	30	24,490	22.8% (95% CI 19.0–26.7)
Group 6	7	35,319	43.3% (95% CI 19.7–66.9)
Group 7	22	107,128	34.2% (95% CI 27.6–40.8)
Group 8	39	18,506	31.0% (95% CI 22.7–39.3)
Group 9	18	43,376	40.7% (95% CI 28.5–52.9)

Group 1: treatment-naïve patients; Group 2: patients receiving curative treatment; Group 3: patients receiving palliative treatment; Group 4: patients receiving curative or palliative treatment, or treatment intent not specified; Group 5: after curative treatment; Group 6: patients without feasible anti-cancer treatment; Group 7: including patients in different phases of treatment; Group 8: patients on anti-cancer treatment; Group 9: patients with advanced, metastatic or terminal disease.

**Table 10 cancers-15-00591-t010:** Meta-regression to explore determinants of pain prevalence.

Determinant	β (95% CI)	*p*-Value
Age (years)		
18–49	0.45 (0.37 to 0.54)	ref.
50–59	0.01 (–0.09 to 0.11)	0.787
60–69	–0.03 (–0.13 to 0.07)	0.570
70+	0.01 (–0.12 to 0.15)	0.843
Type of cancer		
Prostate	0.25 (0.08 to 0.41)	ref.
Head and neck	0.16 (–0.04 to 0.35)	0.116
Lung	0.22 (0.01 to 0.43)	0.039
Breast	0.19 (0.01 to 0.36)	0.036
Gastro-intestinal	0.18 (–0.01 to 0.36)	0.061
Gynaecological	0.11 (–0.08 to 0.30)	0.253
Haematological	0.21 (0.02 to 0.40)	0.034
Ethnicity		
Caucasian	0.38 (0.18 to 0.59)	ref.
Black	0.36 (–0.15 to 0.87)	0.144
Asian	0.17 (–0.12 to 0.45)	0.219
Hispanic	0.39 (–0.01 to 0.78)	0.054
Performance status		
ECOG 2	0.68 (0.43 to 0.94)	ref.
ECOG 0	–0.28 (–0.74 to 0.17)	0.198
ECOG 1	–0.05 (–0.35 to 0.25)	0.733
Data collection method		
Questionnaire patient	0.43 (0.40 to 0.45)	ref.
Interview patient	0.07 (–0.02 to 0.16)	0.106
Medical record	0.19 (0.05 to 0.32)	0.007
Recall period		
Point	0.50 (0.45 to 0.55)	ref.
Week(s)	–0.08 (–0.14 to –0.02)	0.013
Month(s)	–0.18 (–0.28 to –0.07)	0.001
Setting		
Outpatient	0.42 (0.39 to 0.45)	ref.
Inpatient	0.08 (–0.02 to 0.18)	0.115
Palliative care setting	0.16 (0.01 to 0.32)	0.040
Continent of origin		
Europe	0.39 (0.36 to 0.43)	ref.
North America	0.06 (–0.00 to 0.12)	0.068
South America	0.15 (0.01 to 0.29)	0.033
Asia	0.08 (0.01 to 0.14)	0.016
Africa	0.43 (0.25 to 0.62)	0.000
Australia/New Zealand	0.06 (–0.03 to 0.16)	0.193

ECOG = Eastern Cooperative Oncology Group. Bold indicates significant. The item first mentioned is the reference.

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
