# Peer review of "Update on Prevalence of Pain in Patients with Cancer 2022: A Systematic Literature Review and Meta-Analysis"

_cancers, 2023, doi:10.3390/cancers15030591_

Round 1
Reviewer 1 Report
First of all, I commend your effort and agree with you, pain is a major QoL issue as you have rightly said.
Some suggestions:
1) Section on pain prevalence (3.3 - lines 213-226)-I suggest rewrite. Although factually may be correct info, it doesn't read clearly. What you want to tell the reader is that the group after curative treatment and treatment naive report less pain, so it would be better to present your information with this sort of clarity rather than the way it is written.
2) Same comment for section 3.4
3) Group of patients who could not receive any cancer treatment (Group 6), presumably is a group that went in for hospice or supportive care only. I would suspect this group to be far more heterogenous and also, likely to suffer from other comorbidities that precluded palliative treatments. This group deserves more detail- who did you include? what cancers did this group have? what was the most common reason no palliative treatment was possible? For all we know, it was other health conditions and not cancer itself that was the reason for no treatment. Need more clarity.
4) This group 6 is then clubbed with the palliative/ advancer cancer group, which I do not see the reason for. In my opinion, it adds nothing to your study and makes the results less clear
5) Line 284 "In contrary" should either read "Contrary to" or "In contrast." Otherwise, I really like how you have explained why your results differ from previous reports,
6) Limitations of such a meta-analysis, as you have pointed out, are numerous. This same sort of study may be more valuable if not so broad in its scope. For example, if you looked at prevalence of pain in any of your subgroups you have, like patients receiving palliative intent chemo/advanced cancer, or a particular cancer type, eg breast cancer patients, we may be able to reach more firm conclusions. While it is heartening to see your conclusions, given the very heterogenous and broad population chosen, I think we can only say that it appears these would be the conclusions. It is hard to have any certainty about them because the data is too heterogenous. I realize that it is a meta-analysis so understandably will suffer from this caveat, but as your conclusions appear to be at variance with other published literature, my suggestion is to qualify your concluding statements a bit more and add more detail. You already have done a good job of noting your own limitations, just need to put it all together succinctly in your conclusion.
Overall, a very nice read and thanks to the authors for putting so much effort into this very important topic.
Reviewer 2 Report
The article must be corrected because it has some English inaccuracies.
Please check prepositions, punctuation,plural, articles etc.
The conclusions are too general and too short. It would be advisable
to improve the last part.
